# Local processing in neurites of VGluT3-expressing amacrine cells differentially organizes visual information

Jen-Chun Hsiang[1,2], Keith P Johnson[1,2], Linda Madisen[3], Hongkui Zeng[3], Daniel Kerschensteiner[1,4,5,6]*

[1]Department of Ophthalmology and Visual Sciences, Washington University School of Medicine, Saint Louis, United States; [2]Graduate Program in Neuroscience, Washington University School of Medicine, Saint Louis, United States; [3]Allen Institute for Brain Science, Seattle, United States; [4]Department of Neuroscience, Washington University School of Medicine, Saint Louis, United States; [5]Department of Biomedical Engineering, Washington University School of Medicine, Saint Louis, United States; [6]Hope Center for Neurological Disorders, Washington University School of Medicine, Saint Louis, United States

**Abstract** Neurons receive synaptic inputs on extensive neurite arbors. How information is organized across arbors and how local processing in neurites contributes to circuit function is mostly unknown. Here, we used two-photon $Ca^{2+}$ imaging to study visual processing in VGluT3-expressing amacrine cells (VG3-ACs) in the mouse retina. Contrast preferences (ON vs. OFF) varied across VG3-AC arbors depending on the laminar position of neurites, with ON responses preferring larger stimuli than OFF responses. Although arbors of neighboring cells overlap extensively, imaging population activity revealed continuous topographic maps of visual space in the VG3-AC plexus. All VG3-AC neurites responded strongly to object motion, but remained silent during global image motion. Thus, VG3-AC arbors limit vertical and lateral integration of contrast and location information, respectively. We propose that this local processing enables the dense VG3-AC plexus to contribute precise object motion signals to diverse targets without distorting target-specific contrast preferences and spatial receptive fields.
DOI: https://doi.org/10.7554/eLife.31307.001

*For correspondence:
kerschensteinerd@wustl.edu

## Introduction

Neurons receive most of their synaptic input on large intricately branched dendritic arborizations. Traditionally, distributed inputs were thought to be summed linearly at the cell body (*Yuste, 2011*). However, recent studies uncovered extensive local processing and clustered plasticity of synaptic inputs, which enhance the computational power of dendrites (*Grienberger et al., 2015*; *Harvey and Svoboda, 2007*; *Kleindienst et al., 2011*; *London and Häusser, 2005*; *Losonczy et al., 2008*). Although less studied, similar local processing occurs in terminal axon arbors, in which presynaptic inhibition and inhomogeneous distributions of voltage-gated ion channels can diversify the output of a single neuron (*Debanne, 2004*; *Asari and Meister, 2012*).

Amacrine cells (ACs) are a diverse class of interneurons in the retina (*Helmstaedter et al., 2013*; *MacNeil and Masland, 1998*). Most of the approximately 50 AC types lack separate dendrites and axons and receive input and provide output through the same neurites (*Diamond, 2017*). Among the few AC types that have been studied in detail, starburst and A17 ACs are critical for direction selectivity and dim light signaling, respectively (*Grimes et al., 2015*; *Amthor et al., 2002*; *Vlasits et al., 2014*; *Yonehara et al., 2016*; *Yoshida et al., 2001*). The radially symmetric arbors of

starburst ACs receive synaptic input and release neurotransmitters near and far from the soma, respectively (*Ding et al., 2016*; *Vlasits et al., 2016*). In a seminal study, *Euler et al. (2002)* discovered by two-photon $Ca^{2+}$ imaging that the four to six primary neurites of starburst ACs with their daughter branches function as independent centrifugal motion sensors. Similarly, A17 ACs were shown to process converging inputs from rod bipolar cells separately (*Grimes et al., 2010*). For most AC types, however, whether arbors process inputs locally or integrate them globally and what specific stimulus features neurites encode remains unknown.

As in most parts of the nervous system, synaptic communication in the retina occurs in dense neuropils in which arbors of neighboring cells overlap extensively (*Helmstaedter et al., 2013*). Population coding in sensory and motor systems has been studied at the level of cell bodies (*Arnson and Holy, 2013*; *Churchland et al., 2012*; *Leonardo and Meister, 2013*), but how cell-type-specific information is organized in population activity in neuropils has not been explored.

VG3-AC neurites stratify broadly in the center of the inner plexiform layer (IPL) forming a dense plexus in which processes of approximately seven cells overlap at any point (*Haverkamp and Wässle, 2004*; *Johnson et al., 2004*; *Kim et al., 2015*). In somatic patch clamp recordings, VG3-ACs depolarize to light increments (ON) and decrements (OFF) restricted to their receptive field center, but hyperpolarize to large ON and OFF stimuli that include their receptive field surround (*Kim et al., 2015*; *Lee et al., 2014*; *Grimes et al., 2011*). In addition, VG3-ACs depolarize strongly to local object motion but hyperpolarize during global image motion as occurs during eye movements (*Kim et al., 2015*). VG3-ACs are dual transmitter neurons. They provide glutamatergic input to a group of motion-sensitive retinal ganglion cell (RGC) types with diverse contrast and stimulus-size preferences (*Krishnaswamy et al., 2015*; *Kim et al., 2015*; *Lee et al., 2014*), and provide glycinergic input to Suppressed-by-Contrast RGCs (SbC-RGCs), inhibiting selectively responses to small OFF stimuli (*Lee et al., 2016*; *Tien et al., 2016*; *Tien et al., 2015*). Whether VG3-AC neurite arbors process inputs locally or integrate them globally, what stimulus features they encode, and how visual information is organized in the population activity of the VG3-AC plexus to support its varied circuit functions is unknown. Here, we used two-photon $Ca^{2+}$ imaging in a novel transgenic mouse line to address these questions.

## Results and discussion

We crossed VG3-Cre mice to a novel transgenic strain (Ai148) expressing the genetically encoded $Ca^{2+}$ indicator GCaMP6f in a Cre-dependent manner enhanced by tTA-based transcriptional amplification. Staining for VGluT3 confirmed that GCaMP6f labeling in VG3-Cre:Ai148 retinas was mostly restricted to VG3-ACs (*Figure 1—figure supplement 2*) with sparse off-target expression in RGCs (*Grimes et al., 2011*; *Kim et al., 2015*). We imaged GCaMP6f signals in scan fields (33 × 33 μm for *Figures 1*, *2* and *4*; 13 × 100 μm for *Figure 3*) in the IPL of flat-mounted retinas at 9.5 Hz with a pixel density of 4.7 pixels / μm². Recording depths of scan fields were registered by their relative distance to the outer and inner boundaries of the IPL (0–100%) detected by imaging transmitted laser light (*Figure 1—figure supplement 2*). Visual stimulation (385 nm) was spectrally separated from GCaMP6f imaging (excitation: 940 nm, peak emission: 515 nm); and recordings were obtained from the ventral retina, where S-opsin dominates (*Haverkamp et al., 2005*; *Wang et al., 2011*). To objectively identify processing domains of VG3-ACs neurites, we segmented images into functionally distinct regions of interest (ROIs) using a serial clustering procedure (*Figure 1—figure supplement 1*; s. Materials and methods).

In somatic patch clamp recordings, VG3-ACs depolarize to small ON and OFF stimuli (*Lee et al., 2014*; *Kim et al., 2015*; *Grimes et al., 2011*). Somatic $Ca^{2+}$ transients exhibited similar ON-OFF profiles (*Figure 1A and B*). To test how ON and OFF responses are distributed across VG3-AC arbors, we recorded $Ca^{2+}$ transients elicited by contrast steps in a small spot (diameter: 100 μm) at different depths of the IPL (*Figure 1B* and *Video 1*). We quantified contrast preferences by a polarity index, ranging from −1 for pure OFF responses to 1 for pure ON responses (see Materials and methods). Polarity indices varied widely between ROIs (n = 5814, n = 11 mice). The distribution of polarity indices shifted with IPL depth, as neurites in the outer IPL (depths < 40%) responded more strongly to OFF stimuli, and neurites in the inner IPL (depths > 40%) responded more strongly to ON stimuli (*Figure 1C,D*). To make sure that the sparse off-target expression of GCaMP6f in RGCs did not contribute significantly to these results, we imaged signals in the IPL of VG3-Cre:Ai148 mice

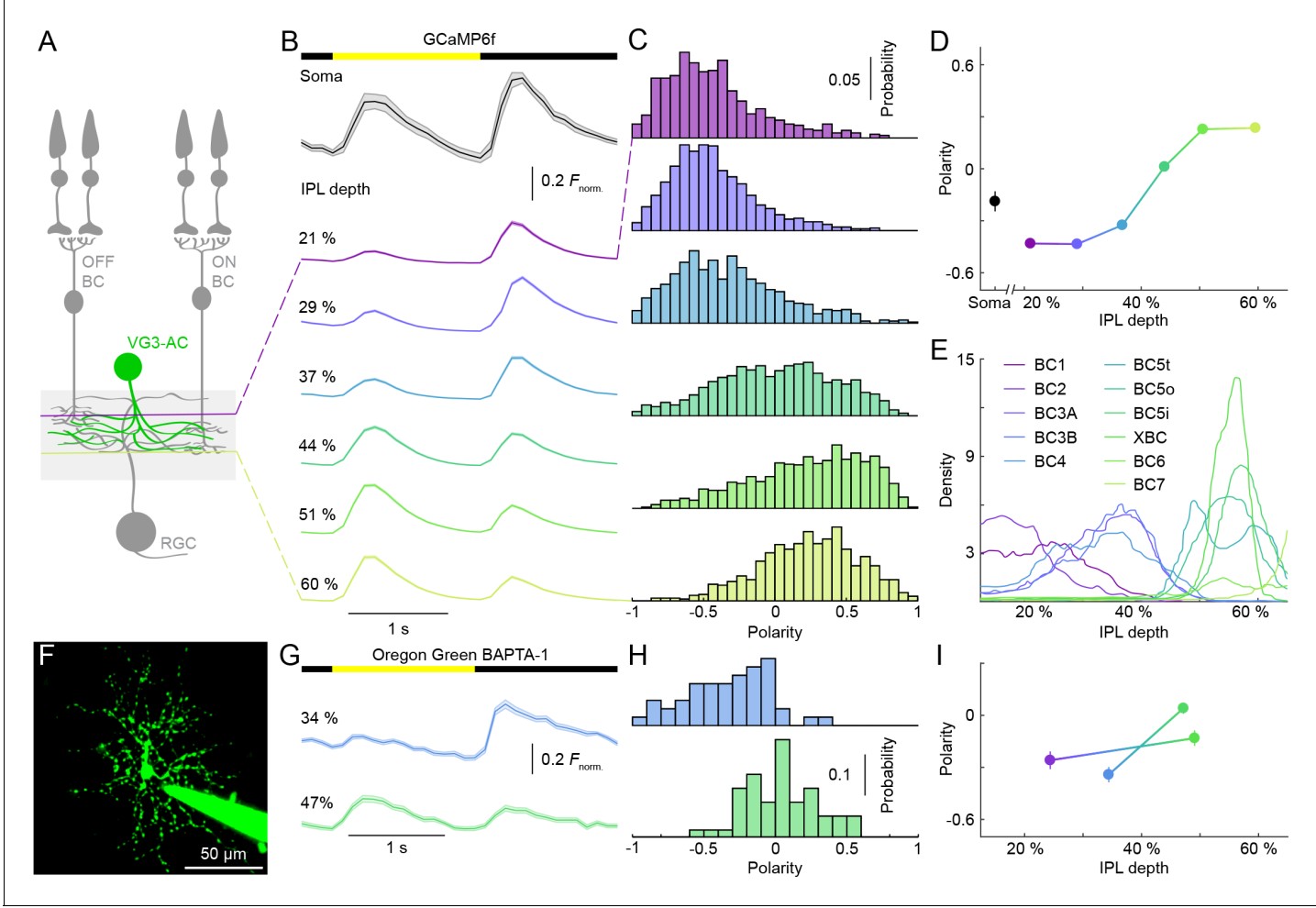

**Figure 1.** Contrast preferences of VG3-AC neurites shift across IPL depths. (A) Schematic of the VG3-AC circuit. VG3-AC neurites receive input from ON and OFF bipolar cells (ON and OFF BC) and synapse onto RGCs. (B) $Ca^{2+}$ transients of ROIs at different imaging depth elicited by contrast steps in a small spot (diameter: 100 μm). A bar at the top indicates the stimulus timing. The black trace (shaded area) shows the mean (±SEM) responses of VG3-AC somata (n = 15). The six color-coded traces (shaded areas) indicate the mean (±SEM) responses of neurite ROIs at different IPL depths (21%: n = 673, purple; 29%: n = 972, blue; 37%: n = 817, sky; 44%: n = 1029, green; 51%: n = 1380, lime; 60%: n = 928, olive). (C, D) Distributions (C) and mean ± SEM (D) of polarity indices of VG3-AC neurite ROIs at different IPL depths color-coded as in (B). Polarity indices differed between IPL depths (p<$10^{-16}$, Kruskal-Wallis one-way ANOVA). ROIs at 21% and 29% IPL depth were more biased to OFF responses than at other depths (p<$10^{-4}$ compared to 37%; p<$10^{-7}$ for 44–60%). ROIs from 51–60% IPL depth were more biased to ON than ROIs from 21–44% (p<$10^{-7}$). No significant differences were observed between 21% and 29% (p=0.99) and between 51% and 60% (p=0.98). Even without image segmentation, using the average activity of each image plane a single data point, polarity indices differed across IPL depths (p<$10^{-12}$, Kruskal-Wallis one-way ANOVA; 21%: n = 15; 29%: n = 18; 37%: n = 14; 44%: n = 16; 51%: n = 23; 60%: n = 20). (E) Lines show the distributions (i.e. skeleton densities) of axons of different OFF (BC1 – BC4) and ON (BC5t – BC7) bipolar cells types from 15–65% IPL depth, according to (*Greene et al., 2016*; *Helmstaedter et al., 2013*) (F) Representative image of a VG3-AC filled with Oregon Green BAPTA-1 via a patch-clamp pipette. (G, H) The average responses (±SEM, G) and polarity index distributions (H) of ROIs of a single VG3-AC at two IPL depths (34%: n = 50, blue; 47%: n = 59, green). (I) Depth-dependent shift in polarity indices (mean ±SEM) of neurite ROIs of two VG3-ACs filled with Oregon Green BAPTA-1 (depth-dependent differences within cells p<$10^{-8}$ and p<0.05).

DOI: https://doi.org/10.7554/eLife.31307.002

The following figure supplements are available for figure 1:

**Figure supplement 1.** Specificity of GCaMP6f expression, VG3-AC neurite $Ca^{2+}$ responses, and functional image segmentation.
DOI: https://doi.org/10.7554/eLife.31307.003
**Figure supplement 2.** Registration of scan fields of functional GCaMP6f imaging to high-resolution image stacks to identify IPL depth.
DOI: https://doi.org/10.7554/eLife.31307.004
**Figure supplement 3.** Depth-dependent shift in contrast preferences in neurites of VG3-Cre:Ai148 mice 3 weeks after optic nerve crush.
DOI: https://doi.org/10.7554/eLife.31307.005

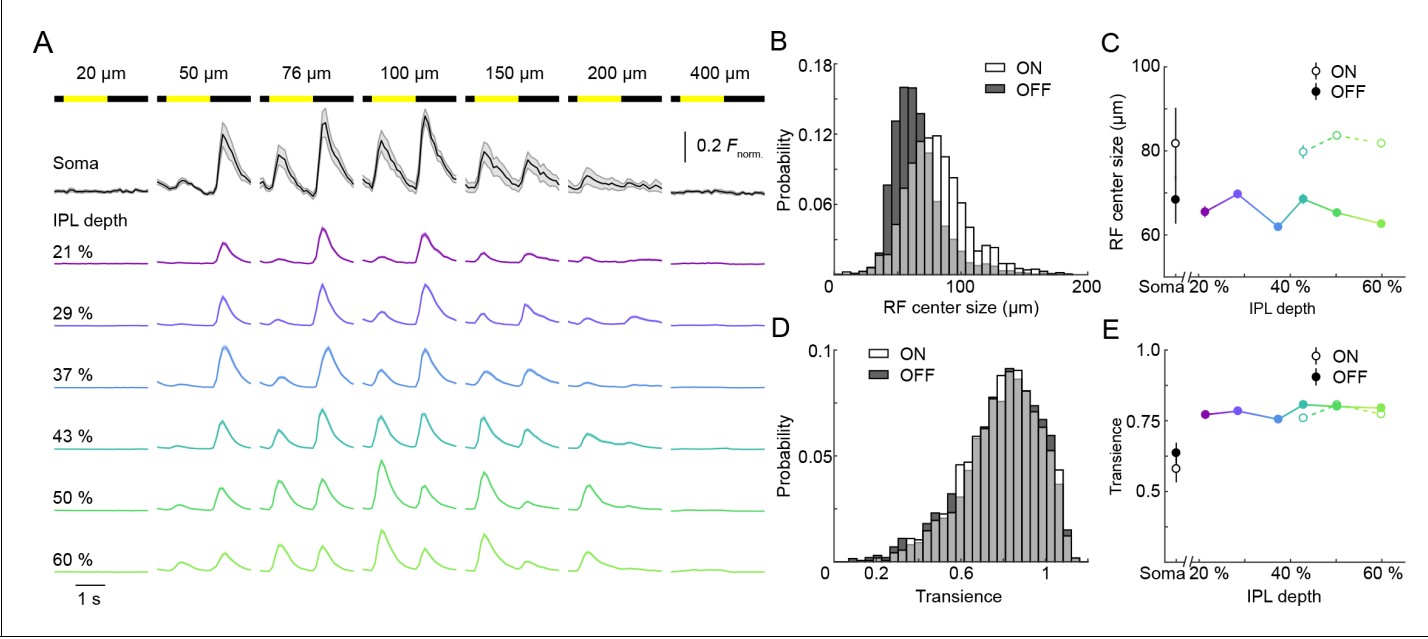

**Figure 2.** ON and OFF responses in VG3-AC neurites differ in preferred stimulus size, but are equally transient. (A) $Ca^{2+}$ responses of ROIs at different imaging depths to contrast steps in spots of different size. Spot diameters are noted above the bars indicating stimulus timing. The black traces (shaded areas) show the mean (±SEM) responses of VG3-AC somata (n = 8). The color-coded traces (shaded areas) indicate the mean (±SEM) responses of ROIs at different IPL depths (21: n = 306, purple; 29%: n = 456, blue; 37%: n = 336, sky; 43%: n = 367, green; 50%: n = 700, lime; 60%: n = 588, olive). (B) The distributions of ON (white) and OFF (dark gray) receptive field center sizes of VG3-AC neurite ROIs. ON receptive field centers were larger than OFF receptive field centers ($p<10^{-99}$, Wilcoxon rank sum test) (C) Receptive field center sizes (mean ±SEM) of ON (open circles) and OFF (filled circles) responses as a function of IPL depth. Because small response amplitudes led to rejection of >50% of ON responses of ROIs from 21–37% IPL depth (s. Material and methods), we restrict comparisons to 43–60% IPL depth. At all these depths, ON receptive field center sizes were larger than OFF receptive field center sizes (43%: $p<10^{-8}$, 50%: $p<10^{-9}$, 60%: $p<10^{-9}$, Wilcoxon rank sum test with multiple comparison correction using the Benjamini–Hochberg procedure). Even without image segmentation, using the average activity of each image plane a single data point, ON receptive field centers were larger than OFF receptive field centers ($p<10^{-3}$, Wilcoxon rank sum test, total: n = 61, 21%: n = 6; 29%: n = 13; 37%: n = 7; 43%: n = 6; 50%: n = 12; 60%: n = 17). (D) The distributions of transience indices of ON (white) and OFF (dark gray) responses of VG3-AC neurite ROIs did not differ significantly (p=0.925, Wilcoxon rank sum test). (E) Summary data (mean ± SEM) of transience indices of ON (open circle) and OFF (filled circle) responses as a function of IPL depth. Due to the high rejection rate of ON responses from 21 to 37% IPL depth, comparisons were restricted to 43–60% IPL depth. Transience indices of ON response is marginally lower than those of OFF responses at 43% (p<0.05), but were not significantly different at 50% (p=0.82) and 60% (p=0.05) IPL depth (Wilcoxon rank sum test with multiple comparison correction using the Benjamini–Hochberg procedure). We are not sure what accounts for the greater response transience observed in VG3-AC neurites vs. somata ($p<10^{-4}$, Wilcoxon rank sum test). One possibility is that inhibitory synaptic inputs favor neurites and abbreviate responses.

DOI: https://doi.org/10.7554/eLife.31307.006

The following figure supplements are available for figure 2:

**Figure supplement 1.** Depth-dependent shift in contrast preferences of VG3-AC neurites is robust across stimulus sizes.
DOI: https://doi.org/10.7554/eLife.31307.007

**Figure supplement 2.** VG3-AC neurites respond selectively to small stimuli.
DOI: https://doi.org/10.7554/eLife.31307.008

**Figure supplement 3.** Scan rates did not limit measurement of VG3-AC neurite response transience.
DOI: https://doi.org/10.7554/eLife.31307.009

3 weeks after optic nerve crush, which removes most RGCs but not ACs (*Park et al., 2008*). Distributions of polarity indices measured in these experiments recapitulated the depth-dependent shift in contrast preferences observed in control retinas (*Figure 1—figure supplement 3*). Because the arbors of each VG3-AC span the depth of the VG3-AC plexus (*Grimes et al., 2011*; *Kim et al., 2015*; *Lee et al., 2014*), it seemed unlikely that the shift in contrast preferences reflected differences between cells. Nonetheless, we imaged $Ca^{2+}$ transients in two VG3-ACs filled with Oregon Green BAPTA-1, confirming that polarity indices shift within arbors of single cells (*Figure 1F–I*). The ratio of ON and OFF signals across VG3-AC arbors closely followed stratification patterns of ON and OFF bipolar cell axons in the IPL (*Figure 1E*) (*Helmstaedter et al., 2013*; *Franke et al., 2017*;

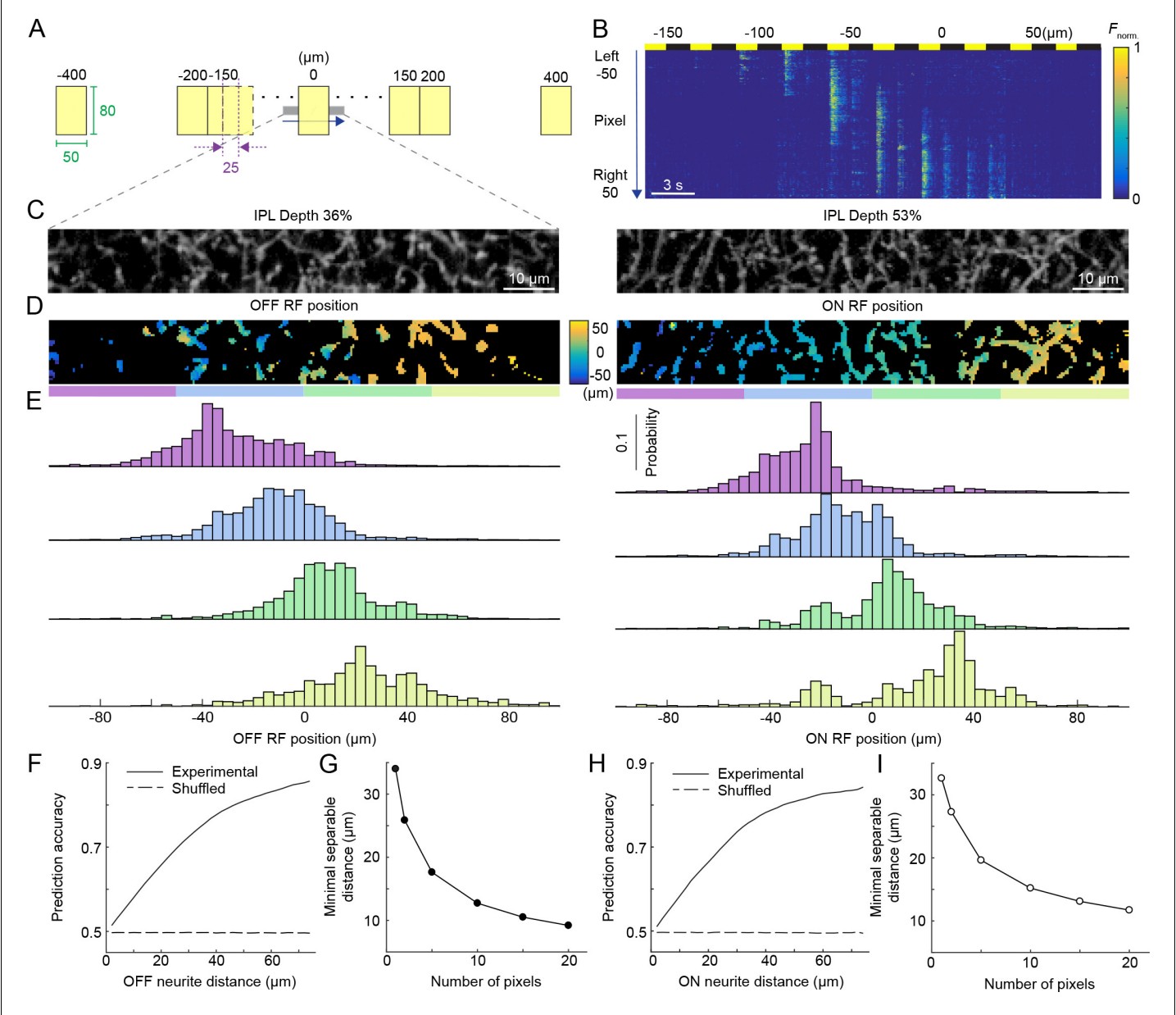

**Figure 3.** Population activity of the VG3-AC plexus encodes spatial information with high precision. (A) Schematic of visual stimulus. Vertical bars (height: 60–80 μm, height: 50 μm width) were presented at 17 different positions along the horizontal axis of a rectangular imaging region (height: 13 μm, width: 100 μm). Stimulus positions were symmetric around the center of the imaging region and spaced by 25 μm center-center distances from −150 μm to 150 μm. In addition, bars were presented −400 μm, −200 μm, 200 μm, and 400 μm from the center of the imaging region. Each bar was presented three for 1.5 s with an interval of 1.5 s between stimuli. The order of stimulus positions was randomized and each stimulus repeated three times. (B) Heatmap of normalized responses in VG3-AC neurites to bars stimuli from −150 μm to 100 μm from the center of the imaging region at an IPL depth of 53%. Responses have been reordered by stimulus positions. Each row of the heatmap represents the activity a single pixel. Pixels were sorted by their distance from the center of the imaging region (−50 μm to 50 μm). (C) Representative images of the VG3-AC plexus in the scan region obtained by averaging the GCaMP6f signal over (left: IPL depth 36%, right: IPL depth 53%). (D) Maps of receptive field positions in the same regions of the VG3-AC plexus shown in (C) (left: OFF responses, right: ON responses). (E) Distributions of receptive field positions of pixels in four adjacent subsections (color-coded from left to right in: purple, sky, lime, and olive) of the scan regions shown in (C) and (D). Receptive field positions of the pixels of each image were aligned to zero their average. (F, H) The accuracy with which a naive Bayes classifier can assign the location of a VG3-AC neurite pixel based on its receptive field position to one of two image subsections increases as a function of the distance between these subsections (solid lines). Dashed lines shows the accuracy when classifiers were trained on shuffled receptive field positions. (G, I) The minimum separable distance (i.e. the point at which prediction accuracy reaches 75%) decreased when predictions were based on multiple pixels (e.g. median ROI size in VG3-AC neurites: 10 pixels).

DOI: https://doi.org/10.7554/eLife.31307.010

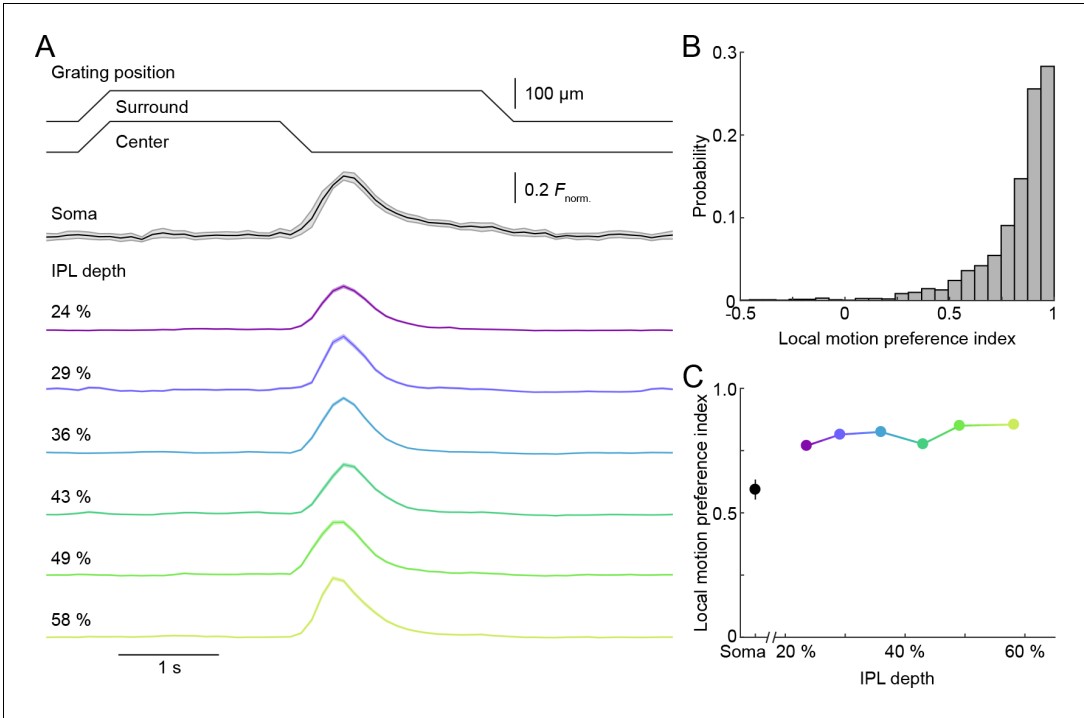

**Figure 4.** Uniform local motion preference of VG3-AC neurites. (**A**) Schematic at the top shows the time course of the grating motion in the receptive field center and surround (s. *Video 2*, and Materials and methods). The black trace (shaded area) shows the mean (±SEM) responses of VG3-AC somata (n = 11). The color-coded traces (shaded areas) indicate the mean (±SEM) responses of ROIs at different IPL depths (24%: n = 388, purple; 29%: n = 202, blue; 36%: n = 500, sky; 43%: n = 322, green; 49%: n = 308, lime; 58%: n = 298, olive). (**B**) The distribution of local motion preference indices of all ROIs. (**C**) Summary data (mean ± SEM) of local motion preference indices as a function of IPL depth. Local motion preference indices did not differ across IPL depths (p=0.09, Kruskal-Wallis one-way ANOVA). No ROI group at any depth was significantly different from any ROI group at another depth.
DOI: https://doi.org/10.7554/eLife.31307.011

The following figure supplement is available for figure 4:

**Figure supplement 1.** Uniform local motion preference of VG3-AC neurites.
DOI: https://doi.org/10.7554/eLife.31307.012

---

*Greene et al., 2016*). However, response polarities of VG3-AC neurites were less extreme than those reported for bipolar cell axons (*Borghuis et al., 2013*; *Franke et al., 2017*). This suggests that local bipolar cell innervation patterns and restricted postsynaptic signal (voltage and/or Ca$^{2+}$) spread determine contrast preferences of VG3-AC neurites and limit vertical integration of visual information in their arbors.

A hallmark of VG3-ACs' somatic voltage responses is strong size selectivity (*Lee et al., 2014*; *Kim and Kerschensteiner, 2017*; *Kim et al., 2015*). We therefore explored how VG3-AC neurites respond to contrast steps in spots of different sizes (*Figure 2A* and *Video 1*). The depth-dependent shift in contrast preferences of VG3-AC neurites observed for 100 μm spots persisted when we calculated polarity indices based on responses to all stimulus sizes (*Figure 2—figure supplement 1*). At all depths, only small stimuli (diameter <400 μm) elicited Ca$^{2+}$ transients in VG3-AC neurites (*Figure 2A* and *Video 1*) and size-selectivity indices of ROIs were uniformly high (*Figure 2—figure supplement 2*), indicating that receptive field surrounds are strong across VG3-AC arbors. To measure ON and OFF receptive field centers, we estimated optimal stimulus sizes for each ROI using a template-matching algorithm (see Materials and methods). ON receptive field centers of VG3-AC neurites were consistently larger than OFF receptive field centers, independent of IPL depth (*Figure 2A–C*). This could be due to larger dendritic territories of the ON compared to the OFF bipolar cells that provide input to VG3-ACs (*Behrens et al., 2016*), and/or the fact that ON but not OFF bipolar cell axons are gap junctionally coupled to AII ACs (*Marc et al., 2014*; *Demb and*

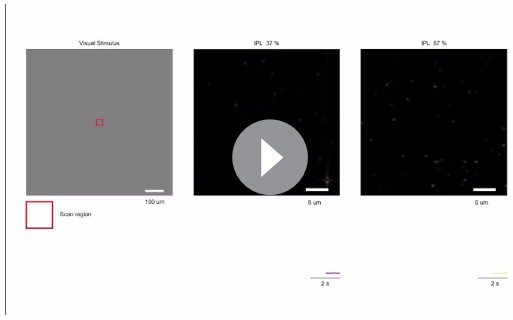

**Video 1.** Ca²⁺imaging of VG3-AC neurite responses to contrast steps in spots of varying size recorded at different IPL depths.
DOI: https://doi.org/10.7554/eLife.31307.013

*Singer, 2015*; *Bloomfield and Völgyi, 2009*). Both ON and OFF receptive field centers were smaller than VG3-AC arbors and only slightly larger than bipolar cell receptive field centers (*Franke et al., 2017*; *Schwartz et al., 2012*; *Purgert and Lukasiewicz, 2015*), supporting the notion that local input from a small number of bipolar cells shapes spatial receptive fields of VG3-AC neurites with limited lateral integration of visual information in their arbors. In contrast to differences in their spatial tuning, ON and OFF responses were equally transient across VG3-AC arbors (*Figure 2A,D,E*). By increasing our scan rate from 9.5 Hz to 37.9 Hz, we confirmed that our measurements of response transience were not limited by the image acquisition rate (*Figure 2—figure supplement 3*).

At any point of the VG3-AC plexus, arbors from approximately seven cells overlap (*Kim et al., 2015*). To explore how spatial information is encoded by population activity in this plexus, we imaged rectangular regions (height: 13 µm, width: 100 µm) in the IPL of VG3-Cre:Ai148 mice while presenting vertically oriented bars (height: 60–80 µm, width: 50 µm) at different positions (interval: 25 µm, range: 800 µm) along the horizontal axis of the imaging region. We presented each bar for 1.5 s with an interval of 1.5 s between subsequent stimuli. Bars were shown in random sequences and responses reordered by stimulus positions in *Figure 3A* and *Video 2*. We analyzed ON and OFF responses separately, but combined data from different IPL depths, which did not differ in their spatial coding (*Figure 2*). For each pixel, we determined receptive field positions along the horizontal stimulus axis (*Figure 3B*; see Materials and methods). This revealed continuous topographic maps of visual space in the VG3-AC plexus (*Figure 3C,D*). To quantify the precision of these maps, we calculated the accuracy with which naive Bayes classifiers could assign neurite activity to specific parts of the map based on receptive field positions (see Materials and methods). Even for single pixels, this accuracy was remarkably high (*Figure 3E,F,H*); and the minimal distance at which different regions of the map could be distinguished with >75% accuracy (i.e. minimal separable distance) decreased further when considering that multiple pixels represent the activity of VG3-AC neurite processing domains (median number of pixels per ROI: 10, *Figure 3G,I*). Thus, local processing generates precise topographic maps of visual space in the population activity of the dense VG3-AC plexus.

VG3-ACs participate in object-motion-sensitive circuits in the retina (*Krishnaswamy et al., 2015*; *Kim et al., 2015*; *Kim and Kerschensteiner, 2017*). We tested the ability of individual VG3-AC neurites to distinguish local and global image motion, using a stimulus in which square wave gratings overlaying center and surround regions of receptive fields moved separately or together (*Kim et al., 2015*; *Olveczky et al., 2003*; *Zhang et al., 2012*). At all IPL depths, isolated motion of the center grating elicited robust Ca²⁺ transients in VG3-AC neurites, which remained silent during simultaneous motion of gratings in center and surround (i.e. global motion) (*Figure 4A*, *Figure 4—figure supplement 1*, and *Video 3*). As a result, local motion preference indices (see Materials and methods) of >70% of ROIs were >0.8 (*Figure 4B,C*). Thus, in spite of the diversity of responses to contrast

**Video 2.** Ca²⁺imaging of VG3-AC neurite responses to white bar at different distance from the center.
DOI: https://doi.org/10.7554/eLife.31307.014

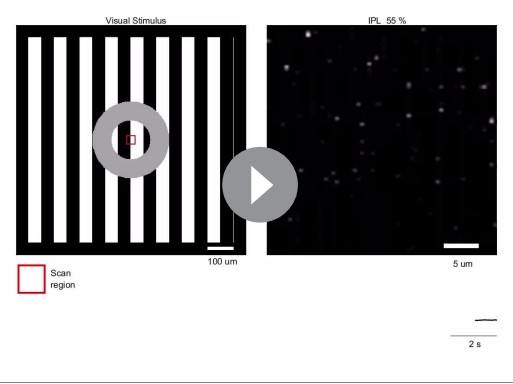

**Video 3.** Ca²⁺imaging of VG3-AC neurite responses to motion.
DOI: https://doi.org/10.7554/eLife.31307.015

steps, VG3-AC neurites exhibit uniform object motion sensitivity.

How does local processing in neurites of VG3-ACs contribute to their circuit function? VG3-ACs provide glutamatergic input to W3-RGCs, which detect movements in a small area of visual space closely aligned with their dendrites (*Kim et al., 2015*; *Zhang et al., 2012*). Input from VG3-ACs is required for normal object-motion-sensitive responses of W3-RGCs (*Kim et al., 2015*). If VG3-ACs integrated visual information globally, excitatory receptive fields of W3-RGCs would expand considerably, lowering the precision with which the position of moving objects could be inferred from their activity (*Jacoby and Schwartz, 2017*). In addition to W3-RGCs, VG3-ACs provide excitatory input to ON direction-selective ganglion cells (ON DSGCs), ON-OFF DSGCs and OFFα-RGCs (*Lee et al., 2014*; *Krishnaswamy et al., 2015*). These motion-sensitive RGC types differ in their preferred stimulus contrast and stratify dendrites at different depths of the IPL. The depth-dependent shift in contrast preferences across neurite arbors likely enables VG3-ACs to contribute motion-sensitive excitatory input to ON DSGCs, ON-OFF DSGCs, and OFFα-RGCs without altering the diverse contrast preference of these targets. VG3-ACs also provide glycinergic input to SbC-RGCs (*Tien et al., 2016*; *Tien et al., 2015*; *Lee et al., 2016*). Whether VG3-ACs release glutamate and glycine from different sites in their arbor and how these sites differ in their visual information remains to be determined. Nonetheless, when VG3-ACs were removed from the retina, inhibitory input to SbC-RGCs was reduced for OFF but not ON stimuli (*Tien et al., 2016*). The local processing of ON and OFF signals we observe in VG3-AC arbors could help explain this selective deficit. Finally, we find that, because of local processing, population activity in the VG3-AC plexus reflects local presynaptic input patterns rather than postsynaptic identity and represents visual space in remarkably precise continuous topographic maps. Thus, local processing enables the dense VG3-AC plexus to contribute precise and uniformly selective object motion signals to diverse targets without distorting target-specific contrast preferences and spatial receptive fields.

## Materials and methods

### Animals

We crossed VG3-Cre mice, in which Cre recombinase is expressed from a bacterial artificial chromosome (BAC) containing regulatory sequences of the *Slc17a8* gene encoding VGluT3, provided by Dr. R.H. Edwards (*Grimes et al., 2011*) to the Ai148 strain, a novel transgenic line made by first targeting a Flp/Frt-based docking site cassette into the TIGRE locus on chromosome 9, followed by modification of that locus by Flp-induced RMCE. Ai148 mice contain Cre-regulated units within the TIGRE locus (*Madisen et al., 2015*) for both GCaMP6f and tTA2 expression, thereby allowing for tTA-based transcriptional amplification of GCaMP6f in a two mouse system. To allow targeting of VG3-ACs under two-photon guidance for filling with Oregon Green BAPTA-1, we crossed VG3-Cre mice to the Ai9 tdTomato reporter strain (*Madisen et al., 2010*). Mice were housed in a 12 hr light/dark cycle and fed *ad libidum*. We isolated retinas from mice of both sexes aged between postnatal day 30 (P30) and P45. All procedures in this study were approved by the Institutional Animal Care and Use Committee of Washington University School of Medicine (Protocol # 20170033) and were performed in compliance with the National Institutes of Health *Guide for the Care and Use of Laboratory Animals*.

## Optic nerve crush

Mice (P30) were anesthetized with a mixture of ketamine (100 mg/kg) and xylene (10 mg/kg). The optic nerve was exposed intraorbitally and crushed with forceps (Dumont #55 FST, Foster City, CA) for ~10 s ~1 mm behind the posterior surface of the eyeball. At the end of surgery, a drop of 0.5% proparacaine hydrochloride ophthalmic solution was administered for pain control, and Melocxican SR (4 mg/Kg) was injected s.c. immediately and 24 hr after surgery. Triple antibiotic ointment (Actavis, Dublin, Ireland) was applied to the cornea for infection prophylaxis.

## Tissue preparation

Mice were dark-adapted for more than 1 hr, deeply anesthetized with $CO_2$, killed by cervical dislocation, and enucleated. Retinas were isolated under infrared illumination in mouse artificial cerebrospinal fluid buffered with HEPES (mACSF$_{HEPES}$ for immunohistochemistry) or sodium bicarbonate (mACSF$_{NaHCO3}$ for two-photon imaging). mACSF$_{HEPES}$ contained (in mM): 119 NaCl, 2.5 KCl, 2.5 $CaCl_2$, 1.3 $MgCl_2$, 1 $NaH_2PO_4$, 11 glucose and 20 HEPES (pH adjusted to 7.37 with NaOH). mACSF$_{NaHCO3}$ contained (in mM) 125 NaCl, 2.5 KCl, 1 $MgCl_2$, 1.25 $NaH_2PO_4$, 2 $CaCl_2$, 20 glucose, 26 $NaHCO_3$ and 0.5 L-Glutamine equilibrated with 95% $O_2$/5% $CO_2$. Isolated retinas were flat mounted on black membrane disks (HABGO1300, MilliporeSigma, Burlington, MA, for immunohistochemistry) or transparent membrane discs (Anodisc 13, Whatman, Maidstone, UK, for two-photon imaging).

## Immunohistochemistry

Flat-mounted retinas were fixed for 30 min in 4% paraformaldehyde in mACSF$_{HEPES}$ at room temperature (RT) and washed three times for 10 min in PBS at RT. The fixed tissue was cryoprotected with incubations in 10%, 20%, and 30% sucrose in PBS for 1 hr at RT, 1 hr at RT, and overnight at 4°C, respectively, followed by three cycles of freezing (held over liquid nitrogen) and thawing (in 30% sucrose in PBS). Retinas were then washed three times in PBS for 1 hr at RT, and stained for VGluT3 (rabbit anti-VGluT3, Cat. No. 1352503, Synaptic Systems, Göttingen, Germany) and GFP (chicken anti-GFP, 1:1000, Cat. No. A10262, ThermoFisher, Waltham, MA) for 3 to 5 days at 4°C in PBS with 5% normal donkey serum and 0.5% Triton X-100. Subsequently, retinas were washed three times for 1 hr in PBS, stained with Alexa 488- Alexa 568-conjugated secondary antibodies (Invitrogen, Carlsbad, CA, 1:1000) overnight at 4°C, washed three times in PBS for 1 hr, and mounted in Vectashield mounting medium (Vector Laboratories, Burlingame, CA) for confocal imaging.

## Confocal imaging

Confocal image stacks of fixed tissue were acquired through 20 × 0.85 NA or 60 × 1.35 NA oil immersion objectives (Olympus, Tokyo, Japan) on an upright microscope (FV1000, Olympus). Confocal images were processed and analyzed with Fiji (*Schindelin et al., 2012*).

## Visual stimulation

Visual stimuli were written in MATLAB (The Mathworks, Natick, MA) using the Cogent Graphics toolbox (John Romaya, Laboratory of Neurobiology at the Wellcome Department of Imaging Neuroscience, University College London, UK). Stimuli were presented from a UV E4500 MKII PLUS II projector illuminated by a 385 nm LED (EKB Technologies, Bat-Yam, Israel) and focused onto the photoreceptors of the ventral retina via a substage condenser of an upright two-photon microscope (Scientifica, Uckfield, UK). All stimuli were centered on the two-photon scan field and their average intensity was kept constant at ~1600 s-opsin isomerizations/S cone/s. To test contrast preferences, receptive field sizes, and response transience, the intensity of spots of varying diameter (20, 50, 76, 100, 150, 200, 400, and 800 µm) was square-wave-modulated (1.5 s ON, 1.5 s OFF) for five cycles. The order in which spots of different size were presented was randomly chosen for each scan field. To probe the distribution of receptive field positions in the VG3-AC plexus, vertical bars (height: 60–80 µm, width: 50 µm) were presented at different positions (interval: 25 µm, range: 800 µm) along the horizontal axis of a rectangular imaging region (height: 13 µm, width: 100 µm). To compare responses to local vs. global motion stimuli, narrow square wave gratings (bar width: 50 µm) over the receptive field center (diameter: 150 µm) and surround (150–800 µm from center of the image) were moved separately or in unison (*Kim et al., 2015*; *Zhang et al., 2012*). A gray annulus was

included in the spatial layout of the stimulus to reliably separate movement in the center and surround. Each grating motion lasted 0.5 s, and movements were separated by 1.5 s.

## Two-photon imaging

A custom-built upright two-photon microscope (Scientifica) controlled by the Scanimage r3.8 MATLAB toolbox was used in this study; and images were acquired via a DAQ NI PCI6110 data acquisition board (National Instruments, Austin, TX). GCaMP6f and Oregon Green BAPTA-1 were excited with a Mai-Tai laser (Spectra-Physics, Santa Clara, CA) tuned to 940 nm, and fluorescence emission was collected via a 60 × 1.0 NA water immersion objective (Olympus) filtered through consecutive 450 nm long-pass (Thorlabs, Newton, NJ) and 513–528 nm band-pass filters (Chroma, Bellows Falls, VT). This blocked visual stimulus light (peak: 385 nm) from reaching the PMT. We compared imaging GCaMP6f signals at higher pixel density (4.7 pixels / $\mu m^2$) and lower scan rate (9.5 Hz), to imaging at lower pixel density (0.85 pixels / $\mu m^2$) and higher scan rate (37.9 Hz). Because image segmentation was more reliable at the higher pixel density and measurements of response transience were indistinguishable between both scan rates (*Figure 2—figure supplement 3*), we acquired images throughout this study at 9.5 Hz with a pixel density of 4.7 pixels / $\mu m^2$. Imaging depths were registered by their relative distances to the borders between the IPL and the inner nuclear layer (IPL depth: 0%) and between the IPL and the ganglion cell layer (IPL depth: 100%). Borders were detected in transmitted light images (*Figure 1—figure supplement 2*). Scan fields at different IPL depths were imaged in pseudorandom order, and for each scan, the retina was allowed to adapt to the laser light for 30 s before presentation of visual stimuli. All images were acquired from the ventral retina, where S-opsin dominates (*Wang et al., 2011*; *Haverkamp et al., 2005*). Throughout the experiments, retinas were perfused at ~7 mL/min with 34°C mACSF$_{NaHCO3}$ equilibrated with 95% $O_2$/5% $CO_2$.

Single VG3-ACs were filled with Oregon Green BAPTA-1 via a patch-clamp electrode in VG3-Cre *Ai9* mice (*Kim et al., 2015*). The intracellular solution contained (in mM): 116 D-gluconic acid (potassium salt), 2 NaCl, 6 KCl, 4 adenosine 5'-triphosphate (magnesium salt), 0.3 guanosine 5'-triphosphate (sodium salt), 20 HEPES, 10 phosphocreatine (disodium salt), 0.15 Oregon Green 488 BAPTA-1. The pH of this solution was adjusted to 7.25 with KOH.

## Image processing

### Registration

Transmitted light images were acquired simultaneously with fluorescence images and were used to detect z-axis displacements that resulted in rejection of the respective image series. Images of series without z-axis displacements were registered to the middle frame using built-in functions in MATLAB. Rigid transformations were applied to both transmitted and fluorescence images. The quality of registration was confirmed by visual inspection, before transformed fluorescence images were used for further image processing and analysis.

### Denoising

Time series of each pixel were searched for outliers (>10 SD). If outliers were isolated in time (i.e. pixel value before and after outlier <10 SD), they were replaced with the average of the value before and after the outlier. This algorithm effectively removed PMT shot noise.

### Segmentation

To identify functional processing domains in VG3-AC neurites with minimal assumptions and user involvement, we developed a serial clustering procedure, in which a functional clustering algorithm is successively applied to different image features. This procedure removed pixels of the image not responding to visual stimulation and automatically assigned responsive pixels to functionally coherent, spatially contiguous regions of interest (ROIs). The functional clustering algorithm was based on *Shekhar et al. (2016)*, beginning with principal components analysis to reduce the dimensionality of the input feature to the minimum needed to explain 80% of its variance. This was followed by a K-nearest-neighbor (KNN) algorithm, which generated a connectivity matrix. The connectivity matrix was then used in community detection clustering (*Le Martelot and Hankin, 2013*). We first applied functional clustering to the raw data of an image series and removed low-intensity pixels. Signals of

remaining pixels were normalized to their peak and fed back into the functional clustering algorithm to group pixels with similar response properties. Groups of functionally similar pixels were divided into spatially contiguous ROIs within the image. The average response traces of these ROIs were subjected to further rounds of functional clustering, in which spatially adjacent ROIs that were grouped in the same cluster were merged. This process was repeated until it converged on a stable solution (typically less than 15 iterations). Finally, ROIs identified in this procedure were examined for signal correlation with the visual stimulus and size, to reject non-responsive and/or small (<5 pixels) ROIs.

To explore encoding of spatial information in the VG3-AC plexus (*Figure 3*), we analyzed distributions of receptive field positions on a pixel-by-pixel basis rather than by ROIs. For this analysis, image series were 2-D median filtered (3 × 3 pixel kernel), and pixels whose standard deviation was in the lower 25% of all pixels were excluded.

## Detrend

To detrend signals from Oregon Green BAPTA-1 imaging, we removed low-frequency fluctuations by ensemble empirical mode decomposition (EEMD) of each ROI (*Wang et al., 2014*). Parameters were set to the following values for EEMD: noise level = 0.1, ensemble number = 100, number of prescribed intrinsic mode functions = 10.

## Image analysis

### Polarity index

Responses of each ROI to contrast steps in small spots (diameter: 100 µm) were divided into ON and OFF periods (1.5 s each, *Figure 1*). The median peak response to five stimulus repeats during each period was then used to calculate a polarity index as follows:

$$Polarity\ index = \frac{Peak_{ON} - Peak_{OFF}}{Peak_{ON} + Peak_{OFF}}$$

A polarity index of 1 indicates pure ON responses, whereas a polarity index of −1 indicates pure OFF responses. To confirm that the observed depth-dependent shift in contrast preferences across VG3-AC arbors was not restricted to a specific stimulus size, we alternatively calculated polarity indices using average responses for all stimulus sizes (diameter: 20–800 µm, *Figure 2—figure supplement 1*).

### Transience index

The transience index (*Figure 2*) was calculated separately for ON and OFF responses of each ROI to contrast steps in its preferred spot size according to:

$$Transience\ index = 1 - \frac{Response\left(t_{peak} + \alpha\right)}{Response\left(t_{peak}\right)}$$

ON and OFF periods each lasted 1.5 s. $t_{peak}$2 is the time to peak, measured from stimulus onset, and $\alpha$2 is a delay set to the fourth frame (~420 ms) after the peak frame. Because response transience was weakly correlated with response amplitude ($R^2$ = 0.0187, p<$10^{-34}$, n = 3631 ROIs), we corrected transience indices by linear regression and rejected responses to ON or OFF stimuli if their maximal amplitude was <25% of the OFF or ON responses of the same ROI, respectively. A corrected maximal transience index of 1.15 indicates that the GCaMP6f signal returned to baseline at time $\alpha$2 after the peak.

### Receptive field center size

Consistent with previous studies (*Crook et al., 2008*), we defined receptive field center size as equivalent to the stimulus size eliciting the maximal response (*Figure 2*). We used a template-fitting algorithm to measure the receptive field center size of each ROI. For each ROI (i.e. target), normalized stimulus-size-response functions of 20 other randomly chosen ROIs (i.e. templates) were scaled and shifted along the x-axis to best fit its own normalized stimulus-size-response. To increase the reliability of curve fitting, stimulus-size-response functions were interpolated from smallest to largest

stimulus size (2 μm intervals) using shape-preserving piecewise cubic interpolation. The receptive field center size of the target ROI was then defined as the average of the estimated optimal stimulus sizes from matching of all 20 template ROIs. Responses to ON or OFF stimuli were rejected if their maximal amplitude was <25% of the OFF or ON responses, respectively.

## Size selectivity index

The peak responses to 100 μm and 400 μm diameter spots of each ROI were used to calculate size selectivity according:

$$Size\ selectivity\ index = \frac{Peak_{100\ \mu m} - Peak_{400\ \mu m}}{Peak_{100\ \mu m} + Peak_{400\ \mu m}}$$

A size selectivity index of 1 indicates the ROI selectively responds to the smaller stimulus (diameter: 100 μm), whereas a index of −1 indicates the ROI selectively responds to the larger stimulus (diameter: 400 μm). ON and OFF responses were analyzed separately.

## Receptive field position and accuracy of location prediction

To analyze how spatial information is encoded in population activity of the VG3-AC plexus (*Figure 3*), we presented vertical bars at different positions along the horizontal axis of a rectangular imaging region (see Visual stimulation). We plotted responses of each pixel as a function of horizontal bar position and fit the relationship with a Gaussian function to estimate the pixel's receptive field position along the horizontal axis. Pixels with receptive field positions > 50 μm outside the image region were rejected (6.9% of all pixels were rejected).

Receptive field positions of pixels in the VG3-AC plexus formed continuous topographic maps. To quantify the precision of these maps, pixels in the image were separated into 38 overlapping bins. Each bin was 25 μm wide, and centers of adjacent bins were 2 μm apart. ON (OFF) responses of pixels were excluded from this analysis if their maximal amplitude was <25% of the OFF (ON) response. For all possible combinations, two bins were selected and assigned to different classes. Pixels from the two bins were randomly split into training and testing sets in 9:1 ratio. Then, a naive Bayes classifier was applied to learn the distributions of receptive field positions in the two bins according to:

$$P(c|x) = \frac{P(x|c)P(c)}{P(x)}$$

where x is the predictor (i.e. the receptive field position of a pixel), $P(x)$ is the prior probability of the predictor, c is the class, $P(c)$ is the prior probability of the class (i.e. the assigned bin of a pixel), $P(x|c)$ is the likelihood of predictor given the class, and $P(c|x)$ is the posterior probability of the class given the predictor. All the probability distributions in the naive Bayes classifier were assumed to be Gaussian distributions. To allow for unbiased estimations with unequal numbers of the pixel from each bin, we resampled the data of each bin to match the bin with the maximum pixel number. Thus, the posterior probability learned by the model directly reflects the likelihood, which is equal to the probability distribution of the receptive field positions in the bin. To measure the accuracy of model predictions, data were split randomly into training and testing sets, and tests performed 100 times. The accuracy of model predictions was then measured as the average of percentage of correct predictions across all 100 splits and tests. To make sure that the model reflects the separation of spatial distributions, we shuffled the receptive field positions between bins for classification, which consistently resulted in the chance level of prediction accuracy. Because VG3-AC neurite processing domains contain more than one pixel (median ROI size: 10 pixels), we tested how prediction accuracy changed when more than one pixel contributes to learning the relationship between image location and receptive field position.

## Local motion preference index

Median responses of each ROI to isolated grating motion in the receptive field center (i.e. local motion) and to synchronous grating motion in receptive field center and surround (i.e. global motion) were used to calculate a local motion preference index as follows:

$$Local\ motion\ preference\ index = \frac{Peak_{Local} - Peak_{Global}}{Peak_{Local} + Peak_{Global}}$$

A local motion preference index of 1 indicates that the respective ROI responded only to local and not to global motion.

### IPL depth sampling

According to previous studies and the GCaMP6f signals in our experiments, neurites of VG3-ACs stratify between 20% and 60% of IPL depth. In our analyses, we binned ROIs into six different depths with equally spaced boundaries from 18% to 62% of IPL depth, encompassing the complete depth of the VG3-AC plexus. In all figures, the depth of each binned data set is given as the average depth of all ROIs within the defined boundaries across all experiments included in the data set.

## Statistics

We acquired functional imaging data from retinas of 17 mice. All summary data and response traces are presented as mean ± SEM. Differences between receptive field center size and transience of ON and OFF responses were statistically examined by Wilcoxon rank sum tests. Tests at different IPL depths were corrected by the Benjamini-Hochberg procedure for multiple comparisons. Depth-dependent differences for polarity and local motion preference indices were tested by Kruskal-Wallis one-way ANOVA, and the paired-group sample-median comparisons were corrected by the Tukey-Kramer method for multiple comparisons.

## Acknowledgements

We thank members of the Kerschensteiner lab and Drs. Josh Morgan and Tim Holy for helpful comments and suggestions throughout this study. This work was supported by the National Institutes of Health (EY023341, EY026978, and EY027411 to DK and the Vision Core Grant EY0268) and by an unrestricted grant from the Research to Prevent Blindness Foundation to the Department of Ophthalmology and Visual Sciences at Washington University.

## Additional information

### Funding

| Funder | Grant reference number | Author |
|---|---|---|
| National Eye Institute | EY023341 | Daniel Kerschensteiner |
| Research to Prevent Blindness | | Daniel Kerschensteiner |
| National Eye Institute | EY026978 | Daniel Kerschensteiner |
| National Eye Institute | EY 027411 | Daniel Kerschensteiner |
| McDonnell International Scholars Academy | | Jen-Chun Hsiang |
| National Institute of General Medical Sciences | GM008151-32 | Keith Johnson |

The funders had no role in study design, data collection and interpretation, or the decision to submit the work for publication.

### Author contributions

Jen-Chun Hsiang, Conceptualization, Software, Formal analysis, Investigation, Writing—original draft, Writing—review and editing; Keith P Johnson, Investigation, Writing—review and editing; Linda Madisen, Hongkui Zeng, Resources, Writing—review and editing; Daniel Kerschensteiner, Conceptualization, Formal analysis, Supervision, Funding acquisition, Writing—original draft, Writing—review and editing

## Author ORCIDs
Hongkui Zeng, http://orcid.org/0000-0002-0326-5878
Daniel Kerschensteiner, http://orcid.org/0000-0002-6794-9056

## Ethics
Animal experimentation: All procedures in this study were approved by the Institutional Animal Care and Use Committee of Washington University School of Medicine (Protocol # 20170033) and were performed in compliance with the National Institutes of Health Guide for the Care and Use of Laboratory Animals.

## Decision letter and Author response
Decision letter https://doi.org/10.7554/eLife.31307.017
Author response https://doi.org/10.7554/eLife.31307.018

## Additional files
### Supplementary files
• Transparent reporting form
DOI: https://doi.org/10.7554/eLife.31307.016

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
