## [Decision Letter]

[Editors’ note: a previous version of this study was rejected after peer review, but the authors submitted for reconsideration. The first decision letter after peer review is shown below.]

Thank you for submitting your work entitled "Local processing of visual information in neurites of VGluT3-expressing amacrine cells" for consideration by *eLife*. Your article has been evaluated by a Senior Editor and three reviewers, one of whom is a member of our Board of Reviewing Editors. The following individual involved in review of your submission has agreed to reveal their identity: Thomas Euler (Reviewer #3).

Our decision has been reached after consultation between the reviewers. Based on these discussions and the individual reviews below, we regret to inform you that your work will not be considered further for publication in *eLife*. We would, however, be happy to reconsider the paper If you can add new data along the lines suggested in the consensus review below.

The reviewers all appreciated the importance of the main question asked here – the degree to which signals are locally processed in neural processes. Both the individual reviews and the discussion among reviewers, however, emphasized that the general significance of the paper could be enhanced considerably, and that this was important for the results to be appreciated by a broad readership. Suggested extensions of the work could be establishing a clearer role for local processing in these amacrine cells in retinal function, or more direct measures of signals in the neurites of individual (presumably sparsely labeled) amacrine cells. More detailed receptive field measurements may help with this. Details on these points are below in the individual reviews.

*Reviewer #1:*

The issue of localized processing in neurons is interesting and important, and is increasingly accessible with advances in genetically-encoded indicators and imaging techniques. This paper exploits these advances to study signals in the neurites of an unusual amacrine cell type. I have no technical concerns about the paper. The experiments appear well done and appropriately controlled and the writing is clear. I do have concerns about the general significance of the work. Specifically, while the paper shows that signals in different neurites can be distinct, it does not tackle either the origin of these differences or their implications for the synaptic output. I think a connection to either inputs or outputs is important for the paper to be appreciated by a broad readership.

Localized processing vs. localized signaling.

The paper has three main results: (1) the ratio of On and Off signals in different neurites differs; (2) different neurites are tuned to different stimulus sizes; (3) neurites are similarly tuned for local vs global motion. The first of these results is expected from the known organization of the inner plexiform layer. The other two could indicate differential and local processing in the amacrine neurites, or could already be present in the inputs to the cell.

Relation to output.

The VGluT-3 amacrines are unusual in using two different transmitters and providing different inputs to downstream targets. The results in the paper are suggestive in this regard – i.e. that localized signaling in the neurites gives rise to local control of the different outputs. Exploring such a connection would greatly enhance the general significance of the paper.

*Reviewer #2:*

Hsiang et al. study the visual response properties of neurite segments of VGluT3-expressing amacrine cells (VG3 ACs) using two-photon calcium imaging. They find that calcium responses in neurites differ as a function of depth within the inner plexiform layer (IPL): neurites that prefer negative contrast stratify nearer to the inner nuclear layer (INL) whereas those that prefer positive contrast stratify further away from the INL. The authors also find that OFF and ON responses differ in their apparent receptive field center diameters at most IPL depths tested. Finally, they find that most VG3 AC neurites respond strongly to local but not global motion stimuli.

The extent to which neurons process stimuli locally or globally is an important outstanding problem. However, this short paper would greatly benefit from additional data to sufficiently support the central claim concerning the locality of integration in VG3 AC neurites.

1) One criticism is that the study is relatively descriptive and does not yield insights into a particular retinal circuit. In this way it differs from the previous studies on starburst amacrine cells and A17 amacrine cells, which yielded specific insights into the direction-selective and rod bipolar cell circuits, respectively.

2) The study would be more powerful if the authors could analyze single VG3-ACs, by sparse genetic labeling, dye loading single cells, etc. This was the method used by Euler et al. and Grimes et al. in their studies of AC dendrites. Otherwise, it is difficult to know whether the distribution of response properties reflect differences between cells or differences between neurites within a cell.

3) One of the main findings is that contrast polarity differs with IPL depth (Figure 2). But this measurement was made with a particular spot size (50-um radius), and the contrast response depends on spot size (Figure 3). The results in Figure 2 would apparently differ depending on the spot size chosen and there might be little effect if the optimal size for ON and OFF response was used in this calculation. The authors should justify their decision to compute contrast preference using a single spot size and/or supplement this analysis by considering the maximal responses to optimal spot sizes.

4) The measurement of center size was difficult to understand and should be illustrated clearly. The data in Figure 3 did not seem to match the example traces in Figure 3 (unless possibly if center size switched from radius to diameter between panels). For example, the ON center size in Figure 3 at 34% depth is ~100, but the response at 100 μm (radius) in Figure 3 is negligible.

*Reviewer #3:*

Hsiang and co-workers studied dendritic signalling in VGluT3 positive amacrine cells in the mouse retina. VGluT3 amacrine cells are unusual as they use glutamate as well as glycine as transmitters. In a series of recent studies – including ones from the Kerschensteiner lab – these amacrine cells were shown to be involved in several retinal circuits, all dealing with different aspects of visual motion. Hsiang et al. utilize a transgenic line in which VGluT3 amacrine cells express a calcium biosensor. Using two-photon calcium imaging, they record light-evoked signals in the dendrites of these cells in different depths of the inner plexiform layer (IPL).

Hsiang and co-workers experimentally confirm what has been proposed based on earlier studies: That the activity in VGluT3 dendrites is highly local, representing the basis for providing signals tailored to the different ganglion cell circuits VGluT3 cells tap into. The authors show that dendritic VGluT3 signals differs in their preference to stimulus polarity (light-ON vs. light-OFF responses), reflecting the IPL depth of the recorded dendritic structures (and the polarity of the respective bipolar cell inputs). In addition, ON and OFF responses differed in their time course (transience) as well as in the stimuli size they preferred. Independent of these differences, all dendritic structured responded consistently to local but not global visual motion. These findings are important as they add a crucial piece to our knowledge about this peculiar amacrine cell; a cell that basically serves as a central "hub" for undirected motion signals in the retina.

This is a very thoroughly conducted study and a well written and illustrated manuscript. I have only a few comments and suggestions.

Results and Discussion, third paragraph: I had problems parsing this section. The authors wrote "variations in response centre size between ROIs were correlated for ON and OFF responses". From the respective figure, I assume that the authors mean that the ratio between preferred stimulus sizes for ON and OFF response components is more or less consistent across the different IPL depths? If so, this could be clarified in the text.

I am surprised that the authors used spots of different sizes and a rather complicated method to extrapolate preferred stimulus size instead of using some kind of noise stimulus and event-triggered averaging to estimate receptive fields. Such an approach may have also allowed to characterize the shape of the receptive fields and the relative position of the ROI within the receptive field. This could tell something about how the dendritic subunits are spatially organised in a particular IPL depth. Have the authors considered/tested this possibility?

[Editors’ note: what now follows is the decision letter after the authors submitted for further consideration.]

Thank you for submitting your article "Local processing in neurites of VGluT3-expressing amacrine cells differentially organizes visual information" for consideration by *eLife*. Your article has been favorably evaluated by Richard Aldrich (Senior Editor) and three reviewers, one of whom, Fred Rieke (Reviewer #1), is a member of our Board of Reviewing Editors. The following individual involved in review of your submission has agreed to reveal their identity: Timm Schubert (Reviewer #3).

The reviewers have discussed the reviews with one another and the Reviewing Editor has drafted this decision to help you prepare a revised submission.

All of the reviewers appreciated the new experiments and analyses and felt this version of the paper was substantially improved over the previous version. The reviewers also all agreed that the paper makes an important contribution to our understanding of neural signaling. There are a few important issues outstanding, as detailed in the individual reviews below.

*Reviewer #1:*

This is a revised version of a paper using calcium imaging to study processing in the neurites of an unusual amacrine cell type. The paper provides interesting insights into the locality of processing in these cells; the study is particularly interesting given the unique role of these cells in retinal processing and their known outputs to several quite different ganglion cell types.

Figure 3 and associated conclusions.

I found several aspects of the experiments and analysis of this figure hard to follow. First, the temporal sequence of the experiment was hard to follow. The text describes the bars as being presented in random sequence. But that made panel B confusing – are these concatenated sections of responses to each bar position? Generally a more complete description of the timing of the data collection would be helpful. Second, the aim of the analysis of positional coding was not clear. One issue would be the precision of the mapping of position to neurites. I think the data collected and the analyses presented are appropriate for that question. Another is the representation of positional information. That question would require information about the noise in the responses that limits the coding accuracy. Thus I do not think the data presented supports the conclusion reached in the fourth and last paragraphs of the Results and Discussion.

*Reviewer #2:*

The authors have performed additional experiments and addressed most of the major concerns with the initial submission. Their major claim, that VGluT3 amacrine cells process signals locally within their dendritic trees, seems to be sound.

With that said, there are a few moderate concerns that could be cleared up.

1) The authors added Ca imaging from two individual cells. The results show some difference in response polarity between inner and outer dendrites, but the difference is certainly small for one of these two cells compared to the population data.

The difference between the two cells, in the On/Off polarity vs. depth analysis, yielded p values that were < 0.05 or < 10^-8. This is a striking difference. It raises a concern with the p < 10^-8 result. This extreme level of significance must be based, in part, on analyzing many small ROIs (yielding a very large n) that are not necessarily independent of one another. This brings up a more significant concern that this process was repeated throughout the paper, leading to the extremely high p values for many of the analyses. I hesitate to bring up such a concern at this point. But it seems necessary to justify the calculation of n (which is linked to the p values) related to the issue of independence between neighboring ROIs in imaging data.

2) It was not clear how the authors confirmed the absence of GCaMP6f+ ganglion cells after the nerve crush procedure. Did they identify a lack of ganglion cell bodies by looking for cells with an axon and not finding them? Or did they stain for ganglion cell markers and show an absence of cells that way?

*Reviewer #3:*

In their manuscript, Jen-Chun Hsiang and colleagues investigate the (subcellular) function of the vGLUT3 amacrine cell, an amacrine cell type that has gained considerable attention in the last five years and plays an important role in the retinal circuits underlying object motion and direction selectivity. The innovative aspect in the work lies in the experimental approach to understand local calcium signaling along the vertical axis of the amacrine cell dendritic tree. This manuscript therefore fills an important gap in our understanding of the underlying mechanisms generating object motion.

To this end, subcellular 2-photon calcium imaging in combination with light stimulation was performed in a transgenic mouse line expressing a calcium biosensor in vGLUT3 cells. They found that ON and OFF signals are confined to separate parts of the dendritic arbor and, therefore, may be relayed to distinct postsynaptic ganglion cells. The manuscript is well written; Data are adequately shown in the figures; the supplemental figures provide background information. Detailed information about sample sizes, p-values and statistical testing is provided.

In particular, a number of control experiments are very thoughtfully conducted and appreciated. For example, optical nerve crush experiments to exclude the contribution of calcium sensors expressed by OFF-ganglion cells and Oregon Green BAPTA 1 measurements to assess whether the contrast preference reflects differences between individual cells. For spectral stimulation, a wavelength of 385nm was used in the ventral, S-opsin dominated retina to avoid bleed through in the GcAMP6 channel.

This reviewer has no major concerns and supports publication.

---

## [Author Response]

[Editors’ note: the author responses to the first round of peer review follow.]

Reviewer #1:The issue of localized processing in neurons is interesting and important, and is increasingly accessible with advances in genetically-encoded indicators and imaging techniques. This paper exploits these advances to study signals in the neurites of an unusual amacrine cell type. I have no technical concerns about the paper. The experiments appear well done and appropriately controlled and the writing is clear. I do have concerns about the general significance of the work. Specifically, while the paper shows that signals in different neurites can be distinct, it does not tackle either the origin of these differences or their implications for the synaptic output. I think a connection to either inputs or outputs is important for the paper to be appreciated by a broad readership.

We thank the reviewer for her/his positive remarks about the importance of the question we addressed and the technical quality of our study. We appreciate the concerns raised about the significance of our findings. To address these concerns and to provide further insights into the origins and implications of local processing of visual information in VG3-AC neurites, we have performed a number of additional experiments and rewritten our manuscript. We outline our new results and changes to the manuscript as they relate to the origins and implications of local visual processing, respectively, in our responses to the following two points.

Localized processing vs. localized signaling.The paper has three main results: (1) the ratio of On and Off signals in different neurites differs; (2) different neurites are tuned to different stimulus sizes; (3) neurites are similarly tuned for local vs global motion. The first of these results is expected from the known organization of the inner plexiform layer. The other two could indicate differential and local processing in the amacrine neurites, or could already be present in the inputs to the cell.

Our revised study has four main results: (1) the ratio of ON and OFF signals varies among VG3-AC neurites as a function of their stratification depth; (2) ON and OFF responses differ in their receptive field sizes independent of stratification depth; (3) signal spread in VG3-AC neurites is limited and stimulus positions are encoded with high accuracy in the population activity of the VG3-AC plexus; (Behrens et al., 2016) VG3-AC neurites show uniform object motion sensitivity. We describe our insights into the origins of these four results below and in the Results and Discussion section our revised manuscript.

Re (1): Because *VG3-Cre* mice sparsely label retinal ganglion cells (RGCs), we wanted to make sure that the observed variation in ON and OFF signals reflected the activity of VG3-ACs. For our revisions, we performed two-photon imaging experiments three weeks after optic nerve crush, which selectively removes a large fraction of RGCs (Park et al. 2008). The ratios of ON and OFF signals in these experiments were indistinguishable from those observed in control retinas, confirming that depth-dependent changes in contrast preferences reflect the activity of VG3-ACs. We show results from optic nerve crush experiments in a new figure supplement (Figure 1—figure supplement 4). Because the arbors of each VG3-AC span the depth of the VG3-AC plexus, it seemed unlikely that the shift in contrast preferences reflected differences between cells. Nonetheless, we filled individual VG3-ACs with the Oregon Green BAPTA-1 via patch clamp electrodes, confirming that polarity indices shift within arbors of single cells. We include results from these experiments in Figure 1 of our revised manuscript. Because the ratio of ON and OFF signals in VG3-AC neurites closely follows the stratification patterns of ON and OFF bipolar cell axons in the inner plexiform layer (IPL), we conclude that local bipolar cell innervation patterns determine contrast preferences of VG3-AC neurites with limited vertical integration of visual information in their arbors.

Re (2): To map receptive fields more precisely than in our initial submission, we performed 13 new imaging experiments, in which we presented additional stimulus sizes and used a template-fitting algorithm (s. Materials and methods) to measure receptive field sizes. The results from these experiments, which are presented in Figure 2 of our revised manuscript, show that ON receptive fields of VG3-AC neurites are consistently larger than OFF receptive fields, independent of stratification depth. Because ON and OFF receptive field sizes differ within the same neurite, we conclude that these differences originate presynaptically, likely reflecting input from different bipolar cell types. Furthermore, both ON and OFF receptive fields were smaller than VG3-AC arbors, indicating that activity in each neurite is dominated by local input from few bipolar cells with limited lateral integration of visual information.

Re (3): As in most parts of the nervous system, synaptic communication of ACs occurs in dense neuropils, in which arbors of neighboring cells overlap extensively. Population coding in sensory and motor systems has been studied at the level of cell bodies, but how information is organized in population activity in neuropils has not been explored. In the IPL, arbors of approximately seven cells overlap at any point of the VG3-AC plexus (Kim et al. 2015). To provide further insights into the origins and significance of local visual processing in VG3-AC neurites, we wanted to determine how spatial information is encoded the population activity of the VG3-AC plexus. If overlapping neurites of different VG3-ACs receive input from different bipolar cells and/or if signals spread far within arbors, the VG3-AC plexus at any point should contain diverse receptive fields. As a result, the accuracy of stimulus-position-encoding in the population activity would be low. On the other hand, if overlapping neurites of different VG3-ACs receive input from the same bipolar cells and information does not spread far within arbors, receptive fields at any point of the VG3-AC plexus should be uniform and stimulus positions should be encoded with high accuracy in the population activity. To distinguish between these possibilities, we imaged rectangular regions (height: 12.8 μm, width: 100 μm) of the IPL in *VG3-Cre Ai148* mice while presenting vertically oriented bars (height: 60-80 μm, width: 50 μm) at different positions (interval: 25 μm, range: 800 μm) along the horizontal axis of the imaging regions. This revealed that receptive fields at any point of the VG3-AC plexus are uniform and that stimulus positions are encoded with high precision in the population activity. Together with findings discussed in Re (1) and Re (2), this argues that information spread within VG3-AC neurites is limited. In addition, it suggests that overlapping neurites of different VG3-ACs receive input from the same bipolar cells. These are, to our knowledge, the first experiments to explore population coding in a neuropil. They reveal that local presynaptic innervation patterns rather than by postsynaptic identity can determine the organization of visual information. The results from these experiments are presented in a new Figure (Figure 3) in our revised manuscript.

Re (Behrens et al., 2016): We observe uniform object motion sensitive responses in all VG3-AC neurites. Previous studies from our group showed that VG3-AC responses to global motion are suppressed by inhibitory input from GABAergic wide-field amacrine cells and that VG3-AC responses to local motion are driven by rectified input from bipolar cells (Kim et al. 2015 and Kim et al. 2017). However, it remained unclear whether responses to local motion require convergence of ON and OFF bipolar cells. Our finding that even VG3-AC neurites with strongly biased polarity (polarity index ~ 1 or ~ -1) respond robustly to local motion reveals that object motion sensitive responses are generated independent of ON-OFF convergence (Figure 4).

Relation to output.The VGluT-3 amacrines are unusual in using two different transmitters and providing different inputs to downstream targets. The results in the paper are suggestive in this regard – i.e. that localized signaling in the neurites gives rise to local control of the different outputs. Exploring such a connection would greatly enhance the general significance of the paper.

Following the reviewer’s suggestion, we have tried in our revisions through additional experiments, analyses, and in rewriting to clarify the implications of local processing of visual information in VG3-AC neurites for their output signals to and the function of different RGC targets. We outline our insights here and describe them in the Results and Discussion section of our revised manuscript.

VG3-ACs form glutamatergic synapses with W3-RGCs (Lee et al. 2014, Kim et al. 2015, and Krishnaswamy et al. 2015). Excitatory input from VG3-ACs is required for object motion sensitive responses of W3-RGCs (Kim et al. 2015). W3-RGCs respond to stimuli in a small area of visual space closely aligned with their dendrites. The arbors of VG3-ACs and W3-RGCs are similar in size. Thus, if VG3-ACs were to integrate visual information globally, the receptive fields of W3-RGCs would extend considerably beyond their own dendrites, lowering the precision with which the positions of moving objects could be inferred from their activity. In imaging experiments performed for our revisions (s. response to previous comment and Figure 3 of our revised manuscript), we find that receptive fields at any point of the VG3-AC plexus are restricted to a narrow region of visual space. This limited lateral integration (i.e. local visual processing), enables the dense VG3-AC plexus (coverage: ~7) to provide spatially precise object motion sensitive excitatory input to W3-RGCs without distorting the receptive fields of this size-selective target.

In addition to W3-RGCs, VG3-ACs provide glutamatergic input to ON direction-selective ganglion cells (ON-DSGCs), ON-OFF DSGCS, and OFFα-RGCs. These motion sensitive RGC types differ in their preferred stimulus contrast and stratify their dendrites accordingly. We find that VG3-AC neurites exhibit different contrast preferences at different IPL depths (Figure 1). This limited vertical integration (i.e. local visual processing), allows VG3-ACs to provide motion sensitive excitatory input to diverse RGC targets without interfering with their characteristic contrast responses.

VG3-ACs form glycinergic synapses with Suppressed-by-Contrast RGCs (SbC-RGCs) (Lee et al. 2016 and Tien et al. 2016). When VG3-ACs were removed from the retina, inhibitory input to SbC-RGCs was reduced selectively for OFF, but not ON stimuli (Tien et al. 2016). The local processing of ON and OFF signals we observe in VG3-AC neurites could account for this selective deficit. We agree with the reviewer that it would be interesting to know if the VG3-AC neurites that provide excitatory and inhibitory output differ in their visual information. We are currently pursing sequential two-photon imaging and electron microscopic circuit reconstruction experiments to address this question. We hope that the reviewer agrees that this effort is beyond the scope of our current study.

Together, our results on local visual processing explain how the dense VG3-AC plexus can distribute spatially precise object motion signals to different RGC types without distorting receptive fields and contrast preferences of these diverse targets. Our findings highlight the importance of studying neuronal computations at a subcellular level and the organization of information at a population level. We hope the reviewer agrees that our findings are of interest to a broad neuroscience audience.

Reviewer #2:[…] The extent to which neurons process stimuli locally or globally is an important outstanding problem. However, this short paper would greatly benefit from additional data to sufficiently support the central claim concerning the locality of integration in VG3 AC neurites.

We thank the reviewer for her/his positive remarks about the importance of the question addressed in our study and its main findings. Following the reviewer’s suggestions, we have performed a series of additional experiments to further test the locality of integration in VG3-AC neurites and to address other specific comments. We will briefly summarize results that support local visual processing here, and described them in more detail in our responses to subsequent comments of the reviewer and in our revised manuscript. (1) For our revisions, we imaged activity in individual VG3-ACs filled with Oregon Green BAPTA-1, confirming that the ratio of ON and OFF responses differs between neurites of a single cell at different IPL depths (i.e. limited vertical integration of visual information). (2) By mapping receptive fields at higher resolution, we find that ON and OFF receptive fields in VG3-AC neurites are smaller than VG3-AC arbors, indicating that activity of each neurite is dominated by local input from few bipolar cells (i.e. limited lateral integration of visual information). (3) At any point of the VG3-AC plexus, neurites from approximately seven cells overlap. To understand how visual information is encoded at a population level, we presented a new stimulus and analyzed the distribution of receptive field positions in the VG3-AC plexus. This revealed continuous topographic maps, which encode stimulus positions with remarkable precision. These maps further supports the notion that lateral integration of visual information is limited and explain how the dense VG3-AC plexus can contribute spatially precise signals to diverse RGC targets without distorting their receptive fields.

1) One criticism is that the study is relatively descriptive and does not yield insights into a particular retinal circuit. In this way it differs from the previous studies on starburst amacrine cells and A17 amacrine cells, which yielded specific insights into the direction-selective and rod bipolar cell circuits, respectively.

We thank the reviewer for this comment. We performed additional experiments to clarify the diversity of contrast processing in VG3-AC neurites (s. responses to points 2 and 3) and to map their receptive fields in more detail (s. responses to points 3 and 4). We also designed a new stimulus to determine the accuracy with which stimulus positions are encoded at population level in the VG3-AC plexus (s. Figure 3). Results from these experiments helped clarify how local visual processing in VG3-AC neurites likely contributes to the function of specific retinal circuits. Here, we briefly outline our insights, which are presented in the Results and Discussion section of our revised manuscript.

VG3-ACs provide excitatory input to W3-RGCs (Lee et al. 2014, Kim et al. 2015, and Krishnaswamy et al. 2015). The input from VG3-ACs is required for object motion sensitive responses of W3-RGCs, which detect movement in a small area of visual space closely aligned with their dendrites (Zhang et al. 2012 and Kim et al. 2015). The arbors of VG3-ACs and W3-RGCs are similar in size. If VG3-ACs were to integrate visual information globally, the receptive fields of W3-RGCs would extend considerably beyond their own dendrites, lowering the precision with which the position of moving stimuli could be inferred from their activity. In imaging experiments performed for our revisions (Figure 3), we find that although arbors of neighboring VG3-ACs overlap extensively (coverage: ~7), receptive field positions in the VG3-AC plexus form continuous remarkably precise maps of visual space. Thus, local processing of visual information (i.e. limited lateral integration), enables the dense VG3-AC plexus to provide object motion sensitive excitatory input to each W3-RGC without distorting its receptive field and without lowering the spatial resolution of object motion detection in the retina.

In addition to W3-RGCs, VG3-ACs provide excitatory input to ON direction-selective ganglion cells (ON DSGCs), ON-OFF DSGCS, and OFFα-RGCs. These motion sensitive RGC types differ in their preferred stimulus contrast and stratify their dendrites accordingly. We find that VG3-AC neurites exhibit different contrast preferences at different IPL depths (Figure 1). Thus, local processing of visual information (i.e. limited vertical integration), allows VG3-ACs to provide motion sensitive excitatory input to diverse RGC targets without interfering with their characteristic contrast coding.

VG3-ACs form glycinergic synapses with Suppressed-by-Contrast RGCs (SbC-RGCs) (Lee et al. 2016 and Tien et al. 2016). When VG3-ACs are removed from the retina, inhibitory input to SbC-RGCs is reduced selectively for OFF, but not ON stimuli (Tien et al. 2016). The local processing of ON and OFF signals in VG3-AC neurites we observe in our imaging experiments could account for this selective deficit.

Together, our findings on local visual processing explain how the dense VG3-AC plexus can contribute spatially precise object motion sensitive signals to diverse RGC types without interfering with target-specific contrast preference and receptive fields.

2) The study would be more powerful if the authors could analyze single VG3-ACs, by sparse genetic labeling, dye loading single cells, etc. This was the method used by Euler et al. and Grimes et al. in their studies of AC dendrites. Otherwise, it is difficult to know whether the distribution of response properties reflect differences between cells or differences between neurites within a cell.

The arbors of each VG3-AC span the depth of the VG3-AC plexus (Grimes et al. 2011, Kim et al. 2015); and the distributions of polarity indices in the outer and inner parts of the plexus show little overlap (s. Figure 1). Thus, is seemed unlikely that the depth-dependent shift in polarity indices reflects differences between VG3-ACs. Nonetheless, following the reviewer’s suggestion and the examples of Euler et al. and Grimes et al., we filled individual VG3-ACs with Oregon Green BAPTA-1 via patch clamp electrodes. Single cell imaging experiments confirmed that the layer-specific differences in response polarity (i.e. ON vs. OFF) observed at a population level in VG3-AC neurites reflect differences between the neurites of individual cells rather than differences between cells. We show the results of single cell imaging experiments in Figure 1 of our revised manuscript.

3) One of the main findings is that contrast polarity differs with IPL depth (Figure 2). But this measurement was made with a particular spot size (50-um radius), and the contrast response depends on spot size (Figure 3). The results in Figure 2 would apparently differ depending on the spot size chosen and there might be little effect if the optimal size for ON and OFF response was used in this calculation. The authors should justify their decision to compute contrast preference using a single spot size and/or supplement this analysis by considering the maximal responses to optimal spot sizes.

We thank the reviewer for raising this important point. To characterize the size preferences of ON and OFF responses in more detail, we performed new imaging experiments in which we presented additional stimulus sizes (s. response to point 4). In this data set, we observed the largest ON responses to spots of 50-μm or 75-μm radius, and the largest OFF responses to spots with 38-μm or 50-μm radius. As a single stimulus size, therefore, spots of 50-μm radius were suited best to illustrate the shift in contrast preferences across VG3-AC arbors with stratification depth (s. Figure 1). However, to make sure that this shift was not restricted to a particular stimulus size, for our revisions, we also calculated polarity indices based on ON and OFF responses across all stimulus sizes. Results from this analysis are shown in a new figure supplement (Figure 2—figure supplement 1) and reveal the same depth-dependent change in contrast preferences observed in responses to spots of 50-μm radius. Note that, following another reviewer’s suggestion, we give the size of stimulus spots and receptive fields in diameter rather than radius in our revised manuscript.

4) The measurement of center size was difficult to understand and should be illustrated clearly. The data in Figure 3 did not seem to match the example traces in Figure 3 (unless possibly if center size switched from radius to diameter between panels). For example, the ON center size in Figure 3 at 34% depth is ~100, but the response at 100 μm (radius) in Figure 3 is negligible.

For our revisions, we performed 13 new imaging experiments, in which we presented additional spot sizes and measured receptive field center sizes using a template-fitting algorithm (s. Materials and methods). For each ROI (i.e. target), we shifted and scaled (x-axis) stimulus-size-response functions of 20 other randomly chosen ROIs (i.e. templates), to fit responses of the target ROI. We then averaged the optimal stimulus sizes from these matched templates and defined this as the receptive field center size of the target ROI. We present results from these experiments in Figure 2 of our revised manuscript. Previous studies similarly used the stimulus size eliciting the maximal response as a measure of the receptive field center size (e.g. Crook et al. 2008). Before settling on this approach, we tried to fit stimulus-size-response functions with Difference-of-Gaussians models. However, these models fit our data poorly, likely because calcium imaging gives a rectified readout of the membrane potential; and large stimuli, which reliably hyperpolarize VG3-ACs (Lee et al. 2014 and Kim et al. 2015), did not cause a measurable decrease in the GCaMP6f baseline fluorescence (Figure 2). To measure the influence of receptive field surrounds, we calculated a size selectivity index comparing responses to small (diameter: 100 μm) and large (diameter: 400 μm) stimuli, similar to previous studies (e.g. Farrow et al. 2011). Both ON and OFF responses were strongly size selective. We show distributions of size selectivity indices in a new figure supplement (Figure 2—figure supplement 2) in our revised manuscript.

Reviewer #3:[…] This is a very thoroughly conducted study and a well written and illustrated manuscript. I have only a few comments and suggestions.Results and Discussion, third paragraph: I had problems parsing this section. The authors wrote "variations in response centre size between ROIs were correlated for ON and OFF responses". From the respective figure, I assume that the authors mean that the ratio between preferred stimulus sizes for ON and OFF response components is more or less consistent across the different IPL depths? If so, this could be clarified in the text.

We have rewritten this section of the manuscript to clarify our results on receptive field sizes. We have performed new imaging experiments in which we presented additional stimulus sizes and used a template-fitting algorithm to measure receptive field sizes (s. response to next point). Our new results confirm, as the reviewer notes, that ON receptive fields of VG3-AC neurites are larger than OFF receptive fields, independent of the stratification depth. The observation of consistent differences between ON and OFF receptive fields in the same neurite suggests that these differences originate presynaptically, likely reflecting input from different bipolar cell types. We show results from the new experiments in Figure 2 and describe the algorithm used to measure receptive field sizes in the Materials and methods section of our revised manuscript.

I am surprised that the authors used spots of different sizes and a rather complicated method to extrapolate preferred stimulus size instead of using some kind of noise stimulus and event-triggered averaging to estimate receptive fields. Such an approach may have also allowed to characterize the shape of the receptive fields and the relative position of the ROI within the receptive field. This could tell something about how the dendritic subunits are spatially organised in a particular IPL depth. Have the authors considered/tested this possibility?

Following the reviewer’s suggestion, we tried to map receptive fields using binary white noise stimuli (Baden et al. 2016 and Franke et al. 2017). However, response-weighted stimulus averaging and related approaches did not render well-defined receptive fields. Two factors likely contributed to this. (1) ON and OFF inputs converge on VG3-AC neurites. We tried to separate ON and OFF influences using variations of approaches others and we previously described (Gollisch et al. 2008, Pearson et al. 2015). While these approaches can work, they require more data and tend to be less robust than averaging is for purely ON or OFF responsive cells. (2) VG3-ACs receive strong surround inhibition driven by rectified receptive field subunits (Lee et al. 2014, Kim et al. 2015). This inhibition is likely recruited by the binary white noise stimulus, suppressing excitatory inputs and the associated calcium transients (s. also the lack of responses to global pattern motion in Figure 4).

For our revisions, we performed 13 new imaging experiments, in which we presented additional spot sizes and measured receptive field center sizes using a template-fitting algorithm (s. Materials and methods). For each ROI (i.e. target), we shifted and scaled (x-axis) stimulus-size-response functions of 20 other randomly chosen ROIs, to fit responses of the target ROI. We then averaged the optimal stimulus sizes from these matched templates and defined this as the receptive field center size of the target ROI. We present results from these experiments in Figure 2 of our revised manuscript. Previous studies similarly used the stimulus size eliciting the maximal response as a measure of the receptive field center size (e.g. Crook et al. 2008). Before settling on this approach, we tried to fit stimulus-size-response functions with Difference-of-Gaussians models. However, these models fit our data poorly, likely because calcium imaging gives a rectified readout of the membrane potential; and large stimuli, which reliably hyperpolarize VG3-ACs (Lee et al. 2014 and Kim et al. 2015), did not cause a measurable decrease in the GCaMP6f baseline fluorescence (Figure 2). To measure the influence of receptive field surrounds, we calculated a size selectivity index comparing responses to small (diameter: 100 μm) and large (diameter: 400 μm) stimuli, similar to previous studies (e.g. Farrow et al. 2011). Both ON and OFF responses were strongly size selective. We show distributions of size selectivity indices in a new figure supplement (Figure 2—figure supplement 2) in our revised manuscript.

Our new measurements of receptive field center sizes show, consistent with our previous results, that in VG3-AC neurites ON receptive fields are larger than OFF receptive fields. Furthermore, both ON and OFF receptive fields are smaller than VG3-AC arbors, indicating that activity of each neurite is dominated by local input from few bipolar cells (i.e. limited lateral integration of visual information).

To address the reviewer’s question how neurite subunits are organized in the VG3-AC plexus, in which neurites from approximately seven cells overlap at any point (Kim et al. 2015), we imaged activity in rectangular regions (height: 12.8 μm, width: 100 μm) while presenting vertically oriented bars (height: 60-80 μm, width: 50 μm) at different positions (interval: 25 μm, range: 800 μm) along the horizontal axis of the imaging region. We found that at any point of the VG3-AC plexus receptive field positions were uniform. This further supports the notion that lateral integration of visual information is limited, suggests that overlapping neurites from different VG3-ACs receive input from the same bipolar cells, and explains how the dense VG3-AC plexus can contribute spatially precise object motion signals to different RGC targets without distorting their receptive fields. Results from these experiments are presented in a new Figure (Figure 3) in our revised manuscript.

[Editors' note: the author responses to the re-review follow.]

Reviewer #1:[…] Figure 3 and associated conclusions.I found several aspects of the experiments and analysis of this figure hard to follow. First, the temporal sequence of the experiment was hard to follow. The text describes the bars as being presented in random sequence. But that made panel B confusing – are these concatenated sections of responses to each bar position? Generally a more complete description of the timing of the data collection would be helpful.

The reviewer is correct. The bars were shown in random sequences and responses reordered by stimulus positions in panel B to more clearly present the salient information. We have added statements clarifying this and provided further information on stimulus timing (duration of bar presentation [1.5 s] and stimulus intervals [1.5 s]) in the Results section and in the legend of Figure 3.

Second, the aim of the analysis of positional coding was not clear. One issue would be the precision of the mapping of position to neurites. I think the data collected and the analyses presented are appropriate for that question. Another is the representation of positional information. That question would require information about the noise in the responses that limits the coding accuracy. Thus I do not think the data presented supports the conclusion reached in the fourth and last paragraphs of the Results and Discussion.

We thank the reviewer for pointing this out. We agree that our analysis captures the precision with which visual space is mapped onto the VG3-AC plexus, but does not allow us to make inferences about the precision with which stimulus positions can be inferred from the activity of the VG3-AC plexus. As suggested, we have rewritten the sentences highlighted by the reviewer and a sentence in the Abstract.

(Original): “Although arbors of neighboring cells overlap extensively, population activity in the VG3-AC plexus encoded stimulus positions with subcellular precision.”

(Modified): “Although arbors of neighboring cells overlap extensively, imaging population activity revealed continuous topographic maps of visual space in the VG3-AC plexus.”

(Original): “Thus, local processing enables the dense VG3 AC plexus to encode spatial information with remarkable precision in its population activity.”

(Modified): “Thus, local processing generates precise topographic maps of visual space in the population activity of the dense VG3 AC plexus.”

(Original): “Finally, we find that because of local processing, population activity in the VG3 AC plexus reflects local presynaptic innervation patterns rather than postsynaptic identity and encodes stimulus positions with subcellular precision.”

(Modified): “Finally, we find that, because of local processing, population activity in the VG3 AC plexus reflects local presynaptic input patterns rather than postsynaptic identity and represents visual space in remarkably precise continuous topographic maps.”

Reviewer #2:[…] With that said, there are a few moderate concerns that could be cleared up.1) The authors added Ca imaging from two individual cells. The results show some difference in response polarity between inner and outer dendrites, but the difference is certainly small for one of these two cells compared to the population data.The difference between the two cells, in the On/Off polarity vs. depth analysis, yielded p values that were < 0.05 or < 10^-8. This is a striking difference. It raises a concern with the p < 10^-8 result. This extreme level of significance must be based, in part, on analyzing many small ROIs (yielding a very large n) that are not necessarily independent of one another. This brings up a more significant concern that this process was repeated throughout the paper, leading to the extremely high p values for many of the analyses. I hesitate to bring up such a concern at this point. But it seems necessary to justify the calculation of n (which is linked to the p values) related to the issue of independence between neighboring ROIs in imaging data.

We thank the reviewer for raising this concern. We believe that the differences we observe in response polarity as a function of IPL depth and in receptive field center sizes of ON and OFF responses reflect true biological differences rather than spurious statistical significance generated by large n’s. The reasons for this are as follows:

1) We observed no differences in local motion preference indices as a function of IPL depth nor between the transience of ON and OFF responses even though these comparisons involved the same n’s.

2) The effect sizes of differences in polarity indices and receptive field center sizes were quite large, and polarity indices at different IPL depths, ON, and OFF receptive field center sizes formed clearly distinct distributions.

3) To further address the reviewer’s comment, we analyzed polarity indices and receptive field center sizes without image segmentation, using the average activity of each imaging plane as a single data point. This greatly reduced n’s. In spite of this, polarity indices and receptive field center sizes remained significantly different across IPL depth and between ON and OFF responses, respectively. We added the statistics of these comparisons, which follow, to the legends of Figure 1 and 2.

Without segmentation, polarity indices differed between IPL depths (p < 10^-12^, Kruskal-Wallis one-way ANOVA). Activity at 21% and 29% IPL depth was more biased to OFF responses than at other depths (p < 0.05 compared to 44%; p < 10^-5^ for 51% – 60%). Activity from 51% – 60% IPL depth was more biased to ON responses than from 21% – 29% (p < 10^-5^), 37% (p <0.01). (21%: n = 15; 29%: n = 18; 37%: n = 14; 44%: n = 16; 51%: n = 23; 60%: n = 20). We added these statistics to the legend of Figure 1.

Without segmentation, ON receptive field centers were larger than OFF receptive field centers (p < 10^-3^, Wilcoxon rank sum test, total: n = 61, 21%: n =6; 29%: n = 13; 37%: n = 7; 43%: n = 6; 50%: n = 12; 60%: n = 17). We added these statistics to the legend of Figure 2.

2) It was not clear how the authors confirmed the absence of GCaMP6f+ ganglion cells after the nerve crush procedure. Did they identify a lack of ganglion cell bodies by looking for cells with an axon and not finding them? Or did they stain for ganglion cell markers and show an absence of cells that way?

The somata of VG3-ACs are localized exclusively in the inner nuclear layer; and all cells labeled by GCaMP6f in the ganglion cell layer of *VG3-Cre Ai148* mice have axons, suggesting that they are RGCs. To assess the loss of RGCs after unilateral optic nerve crush, we compared the number of GCaMP6fpositive somata in the ganglion cell layer of the affected eye three weeks after nerve crush, to the number of GCaMP6f-positive somata in the ganglion cell layer of the other eye. This revealed that optic nerve crush reduced the RGC density by 74%. We describe our analysis and this result in the legend of Figure 1—figure supplement 3.